# Newborn Screening for the Detection of the *TP53* R337H Variant and Surveillance for Early Diagnosis of Pediatric Adrenocortical Tumors: Lessons Learned and Way Forward

**DOI:** 10.3390/cancers13236111

**Published:** 2021-12-03

**Authors:** Karina C. F. Tosin, Edith F. Legal, Mara A. D. Pianovski, Humberto C. Ibañez, Gislaine Custódio, Denise S. Carvalho, Mirna M. O. Figueiredo, Anselmo Hoffmann Filho, Carmem M. C. M. Fiori, Ana Luiza M. Rodrigues, Rosiane G. Mello, Karin R. P. Ogradowski, Ivy Z. S. Parise, Tatiana E. J. Costa, Viviane S. Melanda, Flora M. Watanabe, Denise B. Silva, Heloisa Komechen, Henrique A. Laureano, Edna K. Carboni, Ana P. Kuczynski, Gabriela C. F. Luiz, Leniza Lima, Tiago Tormen, Viviane K. Q. Gerber, Tania H. Anegawa, Sylvio G. A. Avilla, Renata B. Tenório, Elaine L. Mendes, Rayssa D. Fachin Donin, Josiane Souza, Vanessa N. Kozak, Gisele S. Oliveira, Deivid C. Souza, Israel Gomy, Vinicius B. Teixeira, Helena H. L. Borba, Nilton Kiesel Filho, Guilherme A. Parise, Raul C. Ribeiro, Bonald C. Figueiredo

**Affiliations:** 1Departamento de Saúde Coletiva, Federal University of Paraná, Rua Padre Camargo, 260, Centro, Curitiba 80.060-240, PR, Brazil; karinacfraguas@gmail.com (K.C.F.T.); denisecarvalho@ufpr.br (D.S.C.); 2Instituto de Pesquisa Pelé Pequeno Príncipe, Silva Jardim, 1532, Curitiba 80.250-060, PR, Brazil; eamfalcon@gmail.com (E.F.L.); humberto.ibanez@gmail.com (H.C.I.); anselmohoffmannf@gmail.com (A.H.F.); rosiane.mello@fpp.edu.br (R.G.M.); karin.persegona@fpp.edu.br (K.R.P.O.); ivyparise@gmail.com (I.Z.S.P.); heloisakomechen@gmail.com (H.K.); henriquelaureano@outlook.com (H.A.L.); viniciusbiology@hotmail.com (V.B.T.); 3Oncologia Pediátrica, Hospital Erasto Gaertner, R. Dr. Ovande do Amaral, 201, Jardim das Américas, Curitiba 81.520-060, PR, Brazil; mpianovski@erastinho.com.br (M.A.D.P.); analuizademelorodrigues@gmail.com (A.L.M.R.); vanessakosak@hotmail.com (V.N.K.); giselesantosdeoliveira@gmail.com (G.S.O.); deivid.genetica@gmail.com (D.C.S.); 4Centro de Genética Molecular e Pesquisa do Câncer em Crianças (CEGEMPAC-APACN), Avenida Agostinho Leão Jr., 400, Curitiba 80.030-110, PR, Brazil; custodio.gislaine@gmail.com (G.C.); mirnafigueiredo@hotmail.com (M.M.O.F.); rayssadf@icloud.com (R.D.F.D.); gaparise@gmail.com (G.A.P.); 5Hospital do Câncer, UOPECCAN, R. Itaquatiaras, 769, Santo Onofre, Cascavel 85.806-300, PR, Brazil; carmem.fiori@uopeccan.org.br; 6Faculdades Pequeno Príncipe, Av. Iguaçu, 333, Rebouças, Curitiba 80.230-020, PR, Brazil; isgomy@gmail.com; 7Hospital Infantil Joana de Gusmão, R. Rui Barbosa, 152, Agronômica, Florianópolis 88.025-301, SC, Brazil; tatianaeljaick@gmail.com (T.E.J.C.); denisebousfielddasilva@gmail.com (D.B.S.); 8Secretaria do Estado da Saúde do Paraná, R. Piquiri, 170, Rebouças, Curitiba 80.230-140, PR, Brazil; vivianes@sesa.pr.gov.br; 9Hospital Pequeno Príncipe, Silva Jardim, 1532, Curitiba 80.250-060, PR, Brazil; flora.watanabe@hpp.org.br (F.M.W.); edna.kakitani@gmail.com (E.K.C.); anapkuczynski@hotmail.com (A.P.K.); Gabriela.luiz@hpp.org.br (G.C.F.L.); silvio.avila@hpp.org.br (S.G.A.A.); renatabtenorio@gmail.com (R.B.T.); laine_med@yahoo.com.br (E.L.M.); jositwin@gmail.com (J.S.); niltonkiesel@terra.com.br (N.K.F.); 10Oncologia Pediátrica, Hospital de Clínicas da Universidade Federal do Paraná, R. Gen. Carneiro, 181, Alto da Glória, Curitiba 80.060-900, PR, Brazil; lenizacll@hotmail.com (L.L.); tiago@tormen.com (T.T.); 11Departamento de Enfermagem, Universidade Estadual do Centro-Oeste, UNICENTRO, Rua Padre, R. Salvatore Renna, 875-Santa Cruz, Guarapuava 85.015-430, PR, Brazil; vivianekg@yahoo.com.br; 12Oncologia Pediátrica, Campus Universitário, Universidade Estadual de Londrina, Rodovia Celso Garcia Cid—Pr 445 Km 380, Londrina 86.057-970, PR, Brazil; tanegawa@uel.br; 13Departamento de Ciências Farmacêuticas, Federal University of Paraná, Av. Prefeito Lothário Meissner, 632-Jardim Botanico, Curitiba 80.210-170, PR, Brazil; Helena.hlb@gmail.com; 14Leukemia and Lymphoma Division, Department of Oncology, St. Jude Children’s Research Hospital, Memphis, TN 38105, USA

**Keywords:** *TP53* R337H, genetic testing, adrenocortical tumor, neonatal screening, surveillance

## Abstract

**Simple Summary:**

Adrenocortical tumor (ACT) is rare in children and fatal if not detected early. Children who inherit a mutation of the *TP53* gene tend to develop ACT early in life. In the 1990s, scientists revealed that a *TP53* variant (R337H) was frequent in South Brazil. Therefore, the incidence of ACT in children is 20 times higher in this region than in other countries. We reviewed the records of 16 children with ACT treated in a pediatric hospital in Parana state (southern Brazil) and 134 children registered in the state public registry data. We found a high number of cases with advanced disease, leading to an unacceptable number of deaths. These observations contradict newborn R337H screening and surveillance data, showing that surgical intervention in early cases of ACT is associated with a 100% cure. Newborn screening/surveillance should be implemented in regions with a high frequency of the R337H variant.

**Abstract:**

The incidence of pediatric adrenocortical tumors (ACT) is high in southern Brazil due to the founder *TP53* R337H variant. Neonatal screening/surveillance (NSS) for this variant resulted in early ACT detection and improved outcomes. The medical records of children with ACT who did not participate in newborn screening (non-NSS) were reviewed (2012–2018). We compared known prognostic factors between the NSS and non-NSS cohorts and estimated surveillance and treatment costs. Of the 16 non-NSS children with ACT carrying the R337H variant, the disease stages I, II, III, and IV were observed in five, five, one, and five children, respectively. The tumor weight ranged from 22 to 608 g. The 11 NSS children with ACT all had disease stage I and were alive. The median tumor weight, age of diagnosis, and interval between symptoms and diagnosis were 21 g, 1.9 years, and two weeks, respectively, for the NSS cohort and 210 g, 5.2 years, and 15 weeks, respectively, for the non-NSS cohort. The estimated surveillance/screening cost per year of life saved is US$623/patient. NSS is critical for improving the outcome of pediatric ACT in this region. Hence, we strongly advocate for the inclusion of R337H in the state-mandated universal screening and surveillance.

## 1. Introduction

The incidence of pediatric adrenocortical tumors (ACT) is approximately 20 times higher in southern Brazil [1,2,3] than in other regions of the world [4,5,6]. The presence of a founder *TP53* R337H variant in the population accounts for the increased number of cases of ACT, and other pediatric tumors [4,5]. The co-occurrence of germline *TP53* and activating mutations in β-catenin *(CTNNB1)* is rare in pediatric ACT. In a study using two different cohorts and methods (71 pediatric cases of ACT), activating β-catenin mutations (*n* = 13) were detected only in the tumors of individuals with wild-type *TP53* (*n* = 35) and none of those with germline *TP53* mutations (*n* = 36) [6]. The availability of a reliable and inexpensive genetic test to detect the *TP53* R337H heterozygote in the blood makes newborn screening a reasonable approach for the identification of R337H carriers [7]. Neonatal screening/surveillance (NSS) of R337H carriers consisting of close clinical observation, adrenal cortical hormone blood level monitoring, and scheduled imaging studies was associated with an excellent outcome, revealing the efficacy of surveillance for early diagnosis and intervention [1]. The cumulative risk of ACT is approximately 4% in the first decade of life and gradually declines. This cumulative incidence is 25 times higher than that of children with any type of cancer in this age group in the general population [8]. The risk of other cancers, especially choroid plexus carcinoma [7,9], neuroblastoma, and osteosarcoma [9], is slightly higher in children carrying the R337H variant than in the general population [1,7]. Late-onset (adult) tumors are more common than pediatric tumors in R337H-carrier families [10]. Whether the surveillance of ACT would help in the early diagnosis of other pediatric tumors in this age group remains unknown. Furthermore, surveillance is complex and expensive, requiring frequent hospital visits and blood draws for hormonal testing and imaging studies [1]. Although rare, surveillance test results can be false positive or false negative in Li-Fraumeni syndrome. However, it is important to consider this possibility in large longitudinal studies [11]. Indirect effects of intense surveillance on the condition of carriers include pain and discomfort for participants due to frequent blood draws, imaging studies, travel to clinical visits, missing school time, disruption of routine activities, and psychological harm [12].

Since almost all children carrying the R337H variant who develop ACT show conspicuous clinical findings associated with the overproduction of adrenal cortex hormones early in tumor formation, we simplified the surveillance process by focusing on early parental education to recognize the physical changes caused by excess androgen and cortisol, and scheduled periodic telephone interactions with a health agent to provide ongoing education for parents. We reasoned that repeatedly teaching parents to recognize the first clinical signs associated with excess hormones would have a similar efficacy to that of our previous intense surveillance approach [1]. The second neonatal screening followed by a simplified surveillance protocol of 122 families of newborns carrying R337H revealed that this simplified surveillance was also highly effective [2]. Newborn screening for R337H may be continued with pending approval of the government for the inclusion of R337H in the Parana State universal newborn screening panel, followed by this simplified NSS strategy. Given the identified similarities (prevalence of newborns R337H carriers and ACT incidence) in Santa Catarina State [2], the evaluation shifted from Paraná to the federal government. Other southern and southeastern Brazilian states may also have a high prevalence of R337H and incidence of pediatric ACT in the population. For example, the first cluster of pediatric ACT was reported in a charity hospital in the city of Sao Paulo [13]. In another study involving 35,000 newborns in the city of Campinas, Sao Paulo, the prevalence of R337H was 0.21% [14], which is slightly inferior to that in Santa Catarina (0.24%) and Parana state (0.30%) [2]. There has been no systematic analysis of R337H in other Brazilian states or interest in pursuing universal newborn screening. Individuals in these Brazilian states raised concerns associated with the psychological burden for the families of R337H carriers and the potentially high intervention costs. However, early-onset ACT has a unique clinical presentation and natural history in children. When managed at the time of the first clinical signs of virilization or Cushing, the cure rate approaches 100%, whereas death is almost certain in children with advanced stage ACT.

In this study, we described a retrospective analysis of the children with ACT admitted for treatment to the largest children’s hospital in Curitiba, the capital of Paraná. These children were found to carry the R337H variant at the time of ACT diagnosis but did not participate in newborn screening or surveillance (non-NSS). We compared the tumor weight and stage, and interval from the first signs and symptoms to the diagnosis of ACT, according to when the R337H variant had been detected, through the NSS or at the time of ACT diagnosis. We combined the data of both surveillance protocols in the analysis. In addition, we estimated variations in treatment costs according to disease stage using different data sources.

## 2. Materials and Methods

### 2.1. Subjects

The medical records of unrelated children diagnosed with R337H-associated ACT admitted to the Pequeno Principe Hospital were analyzed for age, disease stage, tumor weight, and interval between the initial clinical signs of virilization or Cushing and the diagnosis of ACT. Children with ACT from families known to carry the R337H mutation were excluded. Targeted *TP53* R337H analysis was performed using the polymerase chain reaction-restriction fragment length polymorphism test assay (PCR-RFLP) for R337H [7]. Patients were classified into disease stages I–IV according to their initial and post-surgical features [15]. Stage I patients were managed with surgery alone. Patients with stage III and IV received a combination of cisplatin, doxorubicin, and etoposide (CDE) plus oral mitotane [15,16,17,18]. Some patients with completely resected large stage II tumors received mitotane [15,16,17].

The goal of surveillance of newborns tested positive for the *TP53* R337H variant is to detect early-onset ACT and provide timely intervention. We mainly focused on ACT during the first five years of life because ACT accounts for about 95% of all R337H-associated malignancies in this age group and can be easily suspected. Most importantly, if ACT is detected early, the cure rate is close to 100%. Other tumors such as choroid plexus carcinoma and neuroblastoma are rarely identified. During our first visit with the parents of an infant carrier, we provide a booklet describing the signs and symptoms associated with ACT (virilization and Cushing syndrome signs). The booklet also describes the signs and symptoms of choroid plexus tumors (irritability, altered feedings, seizure, and persistent crying), and neuroblastoma (weight loss, irritability, poor oral intake, abdominal distention and pain, and skin and subcutaneous lesions). However, whether NSS of ACT would be effective in early diagnosis of other pediatric tumors in this age group remains unknown. The parents are also informed of the increased risk of late-onset (adult) cancer associated with this mutation, but do not engage in active surveillance of children above five years or adults. Nonetheless, parents are encouraged to contact our center to discuss any enquiries regarding signs or symptoms that could be suggestive of a malignancy.

The most important contribution of the identification of R337H in newborns is the education of parents on the pattern of inheritance of the variant and the health consequences of carrying R337H. Free testing for the variant is offered to the parents, siblings of newborns, and all relatives of the parental side segregating the R337H. Counseling is focused on the importance of adopting a healthy lifestyle, the risk of developing other pediatric or adult-onset cancers, and information on established preventive routine procedures (Figure 1). Finally, symptomatic individuals who tested positive for R337H are referred to hospitals.

### 2.2. Newborn Screening and Surveillance Database of Pele Pequeno Principe Research Institute/Pequeno Principe Hospital

In 2005–2010 and 2015–2018, two NSS pilot studies were conducted [1,2]. Briefly, the first study involved identification of the R337H variant from neonatal blood (heel prick) and inviting the families of the children who tested positive for the variant to participate in a surveillance program, consisting of regularly occurring visits to outpatient clinics, imaging studies, and monitoring the blood levels of androgens and cortisol. Information about the clinical manifestations of ACT, treatment, and outcome were provided at the time of the newborn screening consent and reporting of the testing results, and during the clinical visits. In the second study, the surveillance protocol was changed to replace frequent contact with the parents with emphasis on the early detection of the signs and symptoms resulting from an excess of dehydroepiandrosterone sulfate (DHEA-S) and cortisol produced by ACT. Instead of frequent outpatient clinical visits and extensive laboratory evaluations, the families were contacted by phone by an advanced practice nurse. Laboratory evaluations were triggered by suspicious clinical features. In the current report, we updated the data on newborn screening and surveillance of the two previously reported studies. The study was approved by the Ethics Committee of Pequeno Príncipe Hospital and National Research Ethics Committee (CAAE number (Curitiba, Paraná state, Brazil, under the ethical codes CAA: 0023.0.208.000-05 (2005), CAAE 0612.0.015.000-08 (2009, 2012, and 2015).

### 2.3. Costs

Estimating the cost of pediatric cancer in Brazil is very complex. We used different sources to estimate the average costs of treating patients with limited or advanced disease. First, we analyzed data from the Brazilian Public Health System (SUS), which is a constitutionally approved universal health system that guarantees free medical access to the population. In this health system, reimbursement is based on fixed costs established by the federal government. However, the reimbursement for pediatric cancer treatment is insufficient. Therefore, it is subsidized by nongovernment foundations. We reviewed and analyzed the DATASUS registry data of patients with ACT less than 18 years of age registered and treated in any of the Parana State public hospitals between 2006 and 2019. Data collected included the number of independent patients, hospital admissions, and disease stage. We estimated government expenses with chemotherapy based on purchase prices (2019 and 2020) listed in the Management System for Procedures, Drugs and Orthoses, Prostheses, and Special Materials of SUS [19] and Medications Market Regulation Chamber [20]. The chemotherapy regimen typically recommended for patients with stage III or IV disease throughout Paraná State is based on the Children’s Oncology Group ARAR0332 [18]. We assumed that patients with stage III or IV disease received a total of eight courses of cisplatin, doxorubicin, and etoposide. We obtained actual reimbursement data from eight patients with advanced-stage disease treated in a pediatric cancer center (Hospital Erasto Gaertner (Curitiba, Brazil)) throughout their clinical course.

### 2.4. Statistical Analysis

The proportion of patients within each characteristic group was compared using Fisher’s exact test, when appropriate. The median tumor weights between the screening and surveillance groups were compared using the Kruskal–Wallis test. The age at ACT diagnosis for different groups was visualized through a transformation of survival curves, called cumulative events. Its distributions were compared through the log-rank test [21]. The marginal (least-squares) means of the staging between the groups were estimated in the emmeans package [22] and the Tukey test was used for the multiple comparison of means. The level of statistical significance was set at *p* < 0.05 and all computations were performed within the R language [23].

## 3. Results

### 3.1. Single Hospital Cohort vs. Newborn Screening and Surveillance Cohorts

Between 2011 and 2019, 16 children with newly diagnosed ACT were admitted to the Pequeno Príncipe Hospital. All of them were heterozygous for the *TP53* R337H variant. The median age of the 16 patients was 4.0 years (range, 0.7–16.1 years). The median interval between the first symptoms noted by the parents and the diagnosis was 15 weeks (range, 3–48 weeks). Five patients had completely resected small tumors without evidence of metastasis. Thus, they were classified as having stage I. Five patients without evidence of microscopic residual microscopic tumor were classified as stage II due to the large tumor sizes. The median tumor weight was 296 g (range, 18–608 g). Finally, six patients had advanced-stage disease, one with stage III and five with stage IV (Table 1). The management of ACT in these cases consisted of surgery only for patients with disease stages I and II. At the discretion of the primary attending, some patients with stage II disease received mitotane. Patients with stage IV disease were individualized and typically received intensive chemotherapy before or after surgery, following the previously reported guidelines [15,17,18].

Of the 171,649 newborns tested in the first screening, 461 (0.27%) newborns and their 238 siblings aged <15 years were found to be carriers during the first newborn screening pilot study (*n* = 699), but only 347 (49.6%) participated in the surveillance program. Of the 42,438 newborns tested in the second screening, 159 (0.37%) newborns were carriers, but only 122 (76.7%) were confirmed to participate and were from Paraná state, and included in the surveillance program. The reasons for the surveillance rejection included personal reasons, preference for private clinics, and being born but not raised in Paraná state, among others. As of June 2021, 11 children who participated in one of the two NSS studies developed ACT. The median age of the 11 patients was 1.8 years (range, two months to 6.2 years). All of them had stage I disease, and the tumor weight ranged from 1 to 54 g (median, 20 g). The interval between endocrine signs and symptoms was less than three weeks in all cases (Table 1). All these patients are alive and disease-free at follow-up period ranging from 0.5 to 14.8 years. None of the patients developed a second malignancy. The 12th stage I ACT from screening was not included in the present analysis because it was a rare R337H/R337H homozygous case of a 9.3-year-old boy, who died after the fifth recurrence. This genotype may occur every other year. Although the observation time of the second pilot study was short, the first three cases of ACT were diagnosed early in the course (<3 weeks), and the tumor weights were 12 g, 21 g, and 54 g, respectively. This observation suggests that both surveillance strategies are similarly effective in detecting early ACT (<100 g). Hence, the results of the two studies were combined for comparative analysis.

Since tumor weight is an independent prognostic indicator in pediatric ACT, we compared the impact of NSS on tumor weight. Among children with ACT from both NSS cohorts, all had tumor weights less than 100 g, whereas only about 35% of those who did not participate in the surveillance (*p* = 0.0005, Appendix A) did. Parents who participated in the first newborn screening but refused to sign consent for surveillance received information about ACT at time of consent for newborn screening and at time of reporting of testing results. The age at diagnosis of the 10 patients ranged from six months to 7.3 years (median, 1.2 years), three patients were diagnosed with stage I disease and one patient with stage IV disease. The remaining patients were classified as having stage II or III disease. The outcome and follow-up data were incomplete for these patients. Appendix A shows the analysis of tumor weights according to NSS. Disease stage is also a critical prognostic indicator in pediatric ACT. Patients with stage I disease have an excellent prognosis, whereas stage IV disease have a dismal prognosis. The impact of stages II and III on outcomes is less established. Surveillance was significantly associated with stage I disease (*p* = 0.0008; Appendix A).

We did not observe a significant difference in age at diagnosis, surveillance, or non-surveillance (*p* = 0.12; Appendix A). In addition, we examined the distribution of disease stage according to age. We noted that in the non-surveillance group, patients with stage I disease were typically diagnosed under two years of age, whereas those with stage IV disease were older than three years of age. Conversely, in those who participated in the surveillance, stage I disease was observed up to six years of age. Age at diagnosis, tumor size, and disease stage can also be influenced by the interval between the first symptom and diagnosis (Appendix A). Our analysis showed a strong association between symptom duration before diagnosis and NSS. In children participating in the surveillance, diagnoses were made within three months from the start of symptoms of virilization and/or Cushing syndrome, whereas the median duration of symptoms in non-participants was 17 months (range, 3–53; *p* < 0.00002). The median tumor weight, age of diagnosis, and interval between symptoms and diagnosis were 21 g, 1.9 years, and <3 weeks, respectively, for the NSS cohort and 210 g, 5.2 years, and 15 weeks, respectively, for the non-NSS cohort.

### 3.2. DATASUS Registry Cohort and Neonatal Screening and Surveillance Costs

Between 2006 and 2019, 134 cases of pediatric ACT were registered in the government DATASUS registry (mean, 11 per year). This number does not include most pediatric ACT cases from private hospitals (approximately 20%) and may fail to register other cases from SUS (public hospitals). The number of admissions for each patient in the DATASUS-covered hospitals varied from 1 to 17. As expected, there was an association between the number of admissions for patients with stage III and IV (Table 2).

Twenty hospital admissions were required for 20 patients with stage I, 32 hospital admissions were required for stage II, whereas 427 admissions were required for 92 children with stage III or IV disease (*p* < 0.00001 for stage I and/or II vs. III and IV). The reasons for multiple hospital admissions include surgeries for metastases or recurrences, chemotherapy cycles, and management of treatment-related toxicity.

The estimated amount paid by SUS to the different hospitals for surgery and chemotherapy agents was US$ 1,713,171 during the study period or about US$ 12,784 per patient. This amount did not include coverage for expenses during hospitalization and those incurred from the loss of parents’ workdays, transportation, meals, and lodging. Furthermore, because 70% of the patients had advanced-stage ACT, the mortality rates and years of life lost were high among these patients (Table 2). To illustrate the burdensome complexity in managing advanced-stage ACT, we analyzed the reimbursed costs of eight patients admitted to a public charity pediatric cancer center in the Paraná state. The amount reimbursed by SUS corresponds to approximately 60% of the actual care cost, which is supplemented by the Hospital Foundation. It is common that the care of patients with metastatic disease at diagnosis, many of whom eventually succumb to the disease, may extend for several years. Therefore, prolonged suffering is frequent during the management of advanced-stage ACT.

The estimated cost of the first neonatal screening in Paraná state followed by surveillance has been reported [1]. Because of the complexity and high cost of this approach, surveillance has been simplified. We estimated that the cost of surveillance featuring only three visits to the hospital and frequent periodic remote contacts with families would substantially reduce costs compared with the previous. Using data from the simplified NSS study, we estimated that the costs for neonatal screening and surveillance per year for the public system of the Paraná state are approximately US$802,880 per year or US$50,180 per patient per year. Without the simplified NSS intervention, the expected cost per patient is at least US$12,784. Thus, the screening/surveillance costs an additional US$37,396 per patient. Considering the life expectancy of these children to be about 60 years, the cost per year of life saved is US$623 (Figure 2).

## 4. Discussion

In this study, we demonstrated that the outcome of children with ACT continues to be dismal in the Paraná state. Between 2011 and 2019, only 30% of the 16 children referred to the largest pediatric hospital in Curitiba, the state capital, had stage I ACT. These data are corroborated by the analysis of 134 pediatric ACT cases registered between 2006 and 2019 in the Parana State Public Database (DATASUS), which includes cases from rural and urban areas, showing that only 15% of cases had stage I ACT. The extent of disease at diagnosis is the single most important variable associated with the outcome of ACT after surgery [24]. Patients with stage I ACT (complete resected tumors weighing < 100 g) have high probability of disease-free survival, which approached 100% for tumors < 50 g [24]. Conversely, patients with metastatic (stage IV) or residual disease after surgery (stage III) have a high rate of disease progression and poor prognosis [24]. Finally, patients who are classified as stage II have increased relapse rates, but can still be cured with additional surgery and intensive chemotherapy [24]. The low rate of cases with stage I disease identified at Pequeno Príncipe Hospital and other hospitals of Paraná state is surprising given the existing knowledge on the biology and presenting signs of this type of pediatric cancer in Paraná and surrounding states [25,26,27,28,29,30,31]. The delay in diagnosis is associated with higher proportions of stages II, III, and IV, and is in part because children with ACT appear to be healthy and energetic in the early phases of tumor development due to the increased somatic growth of these children, which together may confound untrained parents as precocious puberty. These signs are sometimes ignored or not appropriately investigated, mainly due to the parents’ unawareness, usually with a low education level. When overt signs and symptoms associated with the overproduction of androgens (virilization) or cortisol (Cushing syndrome) are noted, the tumor has already progressed beyond stage I. Almost all ACT weighing < 50 g produce hormones that cause clinical manifestations, which are easily observed by trained parents, as documented in the simplified surveillance study [2]. These tumors are often completely resected and eradicated [24]. Moreover, many children with early signs of virilization and Cushing are suspected to have more common endocrinopathies, and may undergo extensive laboratory investigations that might further delay the diagnosis of ACT. Finally, because of the rarity of pediatric ACT, healthcare providers at the point of care do not consider this tumor in the differential diagnosis.

In the late 1990s, the discovery that children with ACT in this geographic region carried a mutation in the *TP53* (R337H variant) [3], which could be detected in the blood by a simple and inexpensive restriction fragment polymorphism enzymatic test [7], created the opportunity for NSS of carriers. A study of newborns who tested positive for R337H at birth and whose parents provided consent for participation in a surveillance program for early detection of ACT showed that all participants who developed ACT had post-surgical stage I disease. Moreover, no recurrence has been noted among members of this cohort, who had been alive for more than 10 years from diagnosis. This observation proved that a small ACT can be eradicated with surgery alone. The *TP53* R337H variant is considered to have low penetrance for cancer [2,31], and in many families, pediatric ACT is the first clinical manifestation of this variant. During the first 17 years of life, ACT accounts for approximately 90% of all cancers in R337H carriers, whereas ACT accounts for only 12% of pediatric cancers in *TP53* carriers with classic Li-Fraumeni syndrome (LFS) [32]. The lifetime risk of cancer is also lower in R337H carriers than in carriers of classic LFS variants [32,33]. Finally, about 80% of ACT survivors carrying classic LFS *TP53* variants develop a secondary cancer within 30 years from the diagnosis of ACT [32], whereas ACT survivors carrying the R337H variant rarely develop secondary cancers during the first four decades of life [10]. Therefore, most children with post-surgical stage I ACT are expected to have medical conditions comparable to those in the general population. However, the cost and complexity of the surveillance precluded the introduction of R337H testing in the Parana state universal newborn screening. The main argument against the implementation of universal NSS was the relatively low cumulative incidence of ACT in R337H carriers (4%), suggesting that approximately 95% of the children undergoing surveillance do not develop ACT. Therefore, the number of imaging studies and repeated blood draws to test for abnormal levels of adrenal cortex hormones was beneficial only for less than 5% of the carriers. A subsequent retrospective analysis of the first surveillance program [1] revealed that the early recognition of the physical changes associated with overproduction of adrenal cortex hormones could substitute for blood levels and imaging studies to identify children with tumors <100 g. Based on these observations, the simplified surveillance program was designed focusing on education and training of parents to recognize early physical changes associated with overproduction of adrenal cortex hormones [2]. The program consisted of three visits at two-month intervals to the clinic for counseling and education, followed by phone contact by healthcare professionals. A feasibility surveillance study of 122 children positive for the R337H variant using this approach detected three cases with clinical findings suspicious for an ACT (two of them in the present study), where all of them were found to have post-surgical stage I disease.

The most compelling reason to implement the universal newborn screening and simplified surveillance for R337H is that without this intervention, the mortality and morbidity of children who develop ACT are unacceptably high. Despite the inclusion of specific information on pediatric ACT and the medical consequences of the founder R337H variant in the curriculum of the local medical schools and pediatric residency programs, there has been no evidence that the frequency of cases with stage I ACT has substantially increased over the past several years. Similarly, among ACT cases in families who agreed to participate in the newborn screening and were informed of the clinical implications of a positive test but declined to participate in the surveillance program, only 30% of children had stage I disease, suggesting that the information was not retained or was incorrectly interpreted in most cases.

The cost–benefit analysis also strongly supports the surveillance program. The estimated cost without surveillance is US$ 12,784 per patient. These estimates are based on the DATASUS registry, which does not consider the costs associated with nonmedical expenses, which can substantially increase the overall cost. A study found that nonmedical expenses accounted for about 46% of the monthly household income of parents from rural areas and 22% of those from urban areas. Out-of-pocket expenses include travel, accommodation (lodging), food, communication, and work disruption [34]. Moreover, the management of advanced-stage ACT is associated with prolonged exposure to toxic chemotherapy and multiple surgeries. As illustrated in Appendix A, many patients without stage I disease are treated for several years, requiring different chemotherapy regimens and surgical procedures. Therefore, without intervention, the potential for loss of life, suffering, and psychological distress for patients and families are substantial. Whether the risk for second cancers is increased in heavily treated patients remains to be determined.

Conversely, surveillance is highly effective in detecting patients with stage I disease, which is highly curable with surgery alone and a short hospital admission. The surgical cost of eradicating the disease is negligible. The nonmedical costs for neonatal screening and infrastructure to run the surveillance are estimated to be US$50,180 per patient over five years of follow-up. Although the monetary cost per patient in the intervention program is higher than the alternative, the number of lives saved and quality life years are much higher among children undergoing surveillance. Considering that these patients would live for at least 60 years with good quality, the cost-effectiveness ratio is US$623 per quality life-year gained. In pediatrics, a medical intervention that costs less than US$50,000 per year of life saved is considered justifiable [35].

The ethical issues that arise regarding the genetic testing and screening of children have been addressed for other diseases [36,37], but they are not yet clear for hereditary cancer [38,39], such as *TP53* R337H in Southern Brazil. Despite advances in genomic research associated with the *TP53* R337H variant [40], with an increased cure rate with genetic testing and different surveillance protocols [1,2], it is necessary to examine the ethics of this matter. The R337H variant is almost always inherited and is associated with early and late-onset cancers [10,40,41]. A newborn positive case triggers a chain of events such as cancer surveillance of the infant and testing of the siblings younger than ten years of age, parents, and other relatives in the affected parental line. To address the psychological and ethical concerns, the parents of a newborn who tested positive are invited for medical visits to discuss the findings, including confirmatory testing, information on the implications of positive testing, genetic counseling, testing relatives, and age-adapted surveillance for carriers (Figure 3). Counseling is focused on the importance of adopting a healthy lifestyle, the types of adolescent and adult neoplasms associated with this mutation, and preventive routine procedures. However, we do not offer systematic surveillance for late-onset cancers beyond being available to discuss with family members any medical event and to facilitate referral to a medical center. Figure 3 summarizes the estimates from the expected 400 R337H-carrier newborns among 155,000 births per year in the state of Paraná. The simplified protocol protects 16 infants and young children with ACT (4%). Parents are encouraged to consider their autonomy in accepting confirmatory testing, surveillance, and expanded family testing. In this context, vulnerabilities and cultural and socioeconomic conditions are considered. The critical points in this process are to save the lives of children younger than five years of age, avoid suffering associated with intensive chemotherapy, and empower the parents through an education program to make informed decisions.

The ethical impact of whether it is worthwhile to pursue non-intense measures to detect ~14–16 stage I ACTs/year among R337H-carrier children younger than five years, or whether we should avoid psychological exposure of ~95% of the unaffected carrier children, remains unclear. It is also worth noting that *TP53* R337H neonatal screening will ultimately disclose one of the parents and all consenting carrier relatives on the same side of the family, and they all should be monitored as a low cancer risk p53 variant [2]. The neonatal test, if mandatory, should also preserve the interest and privacy of the parents, as illustrated in Figure 3, and only they should decide whether to disclose the result and be enrolled in the surveillance program. Article 10 of the International Declaration of Human Genetic Data [42] could also be understood as “a right” of the parents to ignore the neonatal positive R337H results and reject surveillance.

## 5. Conclusions

The high incidence of pediatric ACT in southern Brazil is a consequence of the population’s high frequency of the germline *TP53* R337H variant. The inclusion of R337H in newborn screening is in the best interests of all children born in this geographic region because surveillance of R337H carriers reduces the mortality and suffering of young children who develop ACT. The entire process should have public governance to protect the children as a group and the autonomy of their parents. Therefore, we strongly advocate for the inclusion of R337H in the state-mandated universal newborn screening, surveillance for the children during the years of highest tumor incidence, and education and psychosocial support for the parents of the affected children.

## Figures and Tables

**Figure 1 cancers-13-06111-f001:**
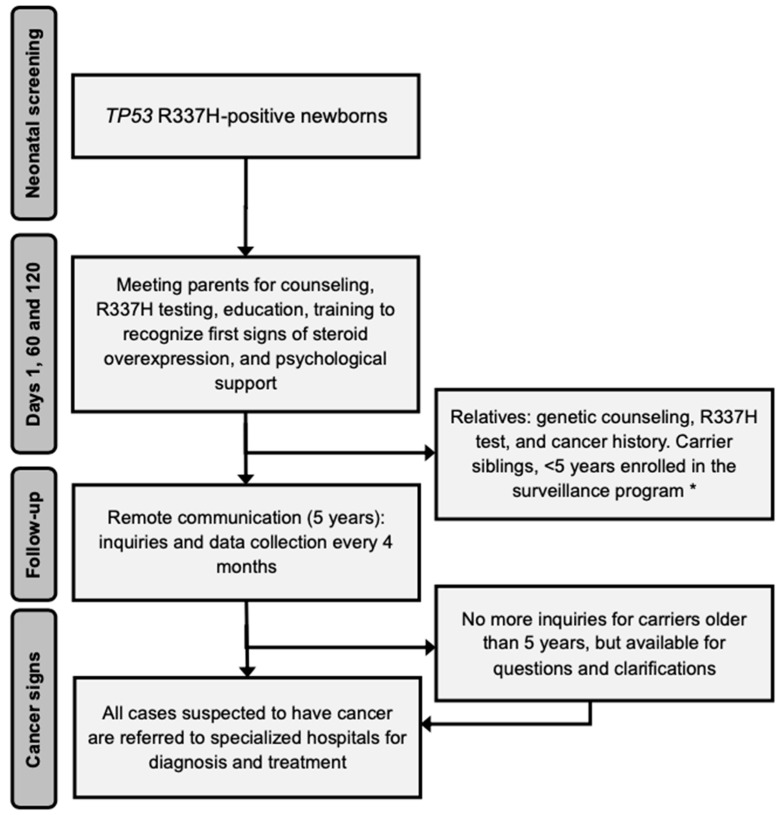
Follow-up flowchart for R337H screening and ACT treatment. * Provided at the study center or at another center closest to the participant home address.

**Figure 2 cancers-13-06111-f002:**
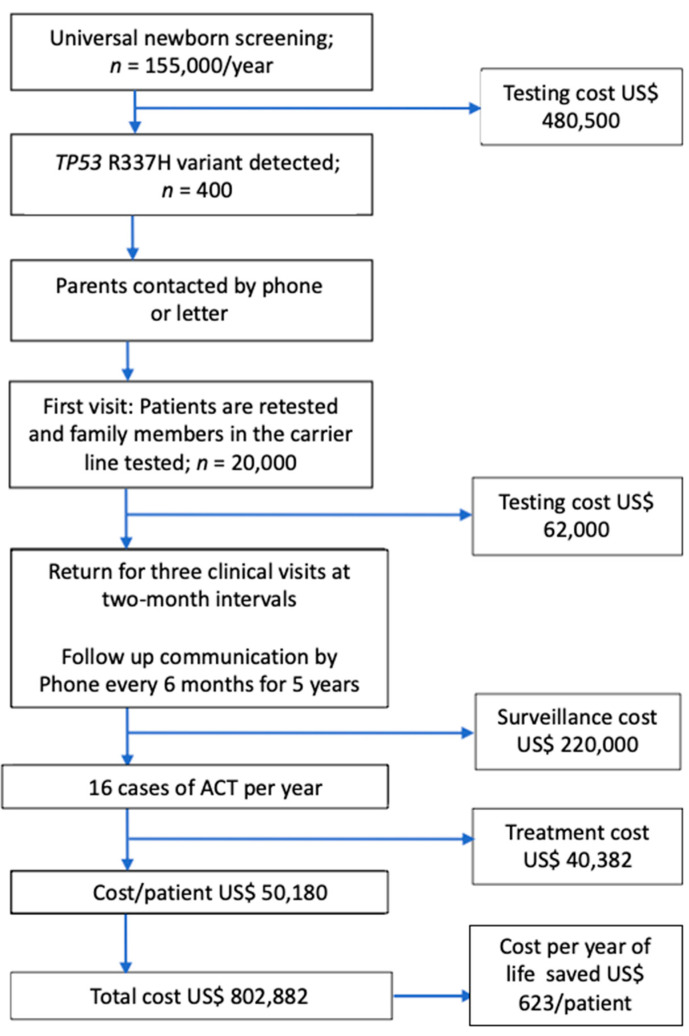
Cost of screening and surveillance. Considering a life expectancy of 60 years, the cost per year of life saved is US$623 per patient.

**Figure 3 cancers-13-06111-f003:**
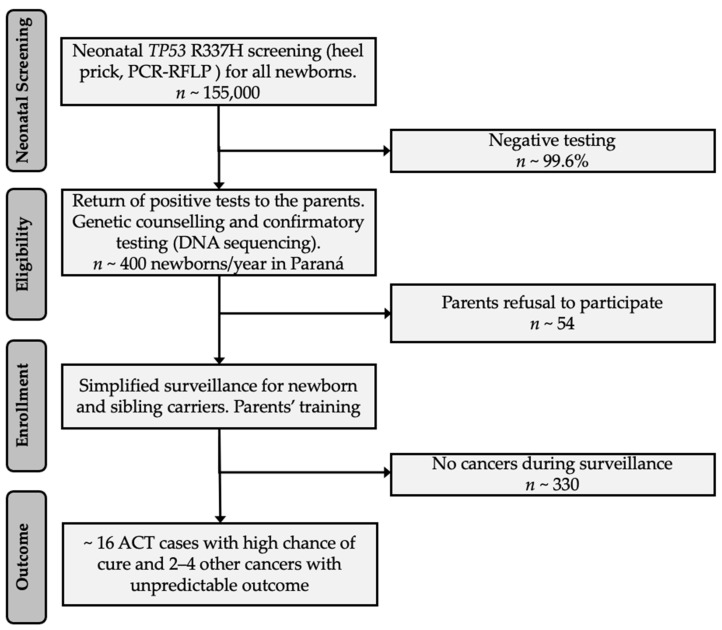
A neonatal and surveillance proposal for the state of Paraná (and state of Santa Catarina with approximately 50% of the projected numbers for Paraná). First step (neonatal screening) expected to be included in the universal Parana and Santa Catarina’s state panel, and provided free of charge. Subsequent steps (eligibility/enrollment) would require the parents’ consent and acceptance to be trained to detect and report early signs and symptoms of ACT.

**Table 1 cancers-13-06111-t001:** Upper: features of cases admitted to a single hospital between 2011 and 2019 (non-NSS); Lower: features of participants in newborn screening and who had tumor detected by surveillance (NSS).

ID	Age at Diagnosis (Years)	Stage	Interval between Symptoms and Diagnosis (Weeks)	Tumor Weight (g)	Treatment
1	2.0	I	32	38	Surgery
2	5.7	I	3	43	Surgery
3	0.7	I	4	18	Surgery
4	1.0	I	12	70	Surgery
5	6.1	I	32	22	Surgery
6	4.0	II	32	318	Surgery/Mitotane
7	4.8	II	24	258	Surgery/Mitotane
8	2.0	II	4	298	Surgery/Mitotane
9	1.0	II	12	126	Surgery
10	2.0	II	24	178	Surgery/Chemo
11	3.1	III	16	376	Surgery/Chemo/Mitotane
12	5.7	IV	8	264	Surgery/Chemo/Mitotane
13	4.0	IV	16	608	Surgery/Chemo
14	7.0	IV	12	242	Surgery/Chemo/Mitotane
15	16.1	IV	40	140	Surgery/Chemo
16	5.0	IV	48	250	Surgery/Chemo/Mitotane
Median	5.2	-	15	210	
1	2.3	I	<3	30	Surgery
2	1.9	I	<3	35	Surgery
3	0.2	I	<3	45	Surgery
4	1.2	I	<3	20	Surgery
5	2.3	I	<3	22	Surgery
6	1.9	I	<3	17	Surgery
7	1.8	I	<3	1	Surgery
8	6.2	I	<3	14	Surgery
9 *	0.9	I	<3	21	Surgery
10 *	2.8	I	<3	54	Surgery
11 *	1.8	I	<3	12	Surgery
Median	1.9	-	<3	21	-

Abbreviations: Chemo, chemotherapy; NSS, newborn screening and surveillance. * Patients participants in the second pilot study; follow-up (years) ranged from 0.5–14.8 years.

**Table 2 cancers-13-06111-t002:** DATASUS registry data of children diagnosed with ACT between 2006 and 2019 admitted to public hospitals in Parana State.

Patients *N* (%)	Disease Stage ^1^	Number of Admissions	Surgery and/or Adjuvant Therapy ^2^ (US$)	Lives Saved ^3^	Years of Life Lost ^4^
20 (14.9%)	I	20	50,460	20	none
22 (16.4%)	II	32	80,763	15	420
92 (68.6%)	III or IV	427	1,581,948	27	3900
134 (100%)	-	479	1,713,171	62	4320

^1^ Staging criteria [15]. ^2^ Costs associated with hospitalizations or other supportive care medications is not included.^3^ Assume survival of 100%, 80% and 30% for patients with disease stage I, II and III/IV, respectively. ^4^ Assuming 60 years of life lost per patient who dies from the disease.

## Data Availability

The datasets used during the current study are available from the corresponding authors upon request.

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
