# Peer review of "Newborn Screening for the Detection of the TP53 R337H Variant and Surveillance for Early Diagnosis of Pediatric Adrenocortical Tumors: Lessons Learned and Way Forward"

_cancers, 2021, doi:10.3390/cancers13236111_

Round 1
Reviewer 1 Report
This is a very manuscript with high impact in the field. However there are several points which should be addressed:
I Please provide data for tumors other than ACC within the positive screen population- which generall preventive routine examination guidelines exist
II Please discuss in which way a positive screening result have an impact for cancer screening beyond ACC
III Please provide a flowchart for follow up screening in patients harboring this special TP 53 mutation
IV please highlight the differences between the TP53 mutation R337H and more common mutations within pTP53
V please highlight if TP53 mutated ACC also displays mutation within beta-catenin
Author Response
We thank you for your thoughtful suggestions and insights.
Reviewer 2 Report
This is a nicely written manuscript that addresses the need for newborn screening and surveillance for early diagnosis of pediatric adrenocortical tumors. The manuscript needs improvement in data presentation and conclusion:
Abstract: The authors need to provide the tumor weight, age at presentation and symptom duration for the newborn screen-surveillance cohort (NSS) in comparison to the non-NSS regular patients. Similarly the cost of treatment of both groups need to be compared
Materials and methods:
2.1 Subjects. This should be labeled clearly as control subjects and preferably placed as 2.2. Newborn screening and surveillance database should be placed first.
2.3. Costs: Why not add the treatment cost estimate of the control non-NSS patients in comparison to the NSS patients who developed ACT? This seems logical in addition to cost estimates from DATASUS
Results:
3.1/3.2 To highlight and compare presentation and symptom duration for the NSS group vs the non-NSS group, these two sections should be combined and the data to present a direct comparison between the two groups.
Tables 1 &2 can be combined into one table to compare the clinical characteristics of the two groups.
3.3/3.4. In a similar fashion, the cost data can be combined to reflect direct comparison of costs between the NSS and SUS groups
Supplementary fig 2 should be placed as regular figure
Conclusion: It is unclear from the stated conclusion whether the authors advocate for newborn screen/surveillance or not. The conclusion should vastly be revised to reflect the study results.
Author Response

(The authors gave the same response as above.)

Reviewer 3 Report
I found the manuscript interesting and well written.
The authots need to clarify in the introduction that the presence of the R337H variant does not implicate that the child will develop ACT. In fact, only a minority of the children with this variant of TP53 will develop ACT.
The authors need to make clear that the results combined the 2 screnings performed - this information need to be grasped since the beginning, and it is not.
In table 1, the "not available information" on the treatment type should be
explained why couldn't be available.
On line 213/page 5 the % of newbornswho are carriers should be stated. Also, on line 251, the prevalence should be showed based on the total of children screened; it is not totally meaningful to state that 11 children develop ACT.
On page 8, line 290 the authors refer to poor quality of life. The study did not assess QoL but the authors may try to detail better this declaration.
On line 295, it is said that 3 visits to the hospital are done. Are 3 per year? It should be detail.
In the Discussion, on line 320 and 321 the information is not clear (ressected without remission).
On line 345 the number of phone calls should be clear. The positive cases detected with the second surveillance is yet limited to reason the benefit of the screenings. Despite the lower numbers, the benefits should be better explain; perhaps with bullets or in a figure.
The number of families with a positive screening but who deny to participate in the surveillance skeme should be discussed as well as ways to decrease this number.
The information on line 384 is conflicting with the notion that children with this TP53 variant live a normal life regarding the risk of having other malignancies.
Nice discussion on the ethics perspectives - line 404-411.
Author Response

(The authors gave the same response as above.)

Round 2
Reviewer 1 Report
Trank you for you reply– I Wolldecke support the publication in Thais Form.
Reviewer 2 Report
The authors have adequately responded to my comments and requests